# Research on Computer-Aided Diagnosis Method Based on Symptom Filtering and Weighted Network

**DOI:** 10.3390/e24070931

**Published:** 2022-07-05

**Authors:** Xiaoxi Huang, Haoxin Wang

**Affiliations:** School of Computer Science and Technology, Hangzhou Dianzi University, Hangzhou 310000, China; huangxx@hdu.edu.cn

**Keywords:** computer-aided diagnosis, symptom filtering, weighted network, hierarchical reinforcement learning

## Abstract

In the process of disease identification, as the number of diseases increases, the collection of both diseases and symptoms becomes larger. However, existing computer-aided diagnosis systems do not completely solve the dimensional disaster caused by the increasing data set. To address the above problems, we propose methods of using symptom filtering and a weighted network with the goal of deeper processing of the collected symptom information. Symptom filtering is similar to a filter in signal transmission, which can filter the collected symptom information, further reduce the dimensional space of the system, and make the important symptoms more prominent. The weighted network, on the other hand, mines deeper disease information by modeling the channels of symptom information, amplifying important information, and suppressing unimportant information. Compared with existing hierarchical reinforcement learning models, the feature extraction methods proposed in this paper can help existing models improve their accuracy by more than 10%.

## 1. Introduction

With the development of machine learning and deep learning, artificial intelligence technologies are being used to varying degrees in all corners of society, including the medical field. Artificial intelligence not only provides tools to support hospitals and patients, but also creates a new healthcare ecosystem with new ways of engagement, new modes of interaction, and new interrelationships [1]. Many technicians have previously attempted to use machine learning or deep learning to complete the computer-aided diagnosis of before. L Li et al., completed the identification of type 2 diabetes [2], Edward Choi et al., used GRU to model the time series relationship of patient health records to diagnose premature heart failure [3], and Yining Zhang et al., used perceptron machine learning algorithms to analyze heart disease-related data, and finally came to a judgment of whether the person had heart disease [4]. Qiang Chen has implemented a buffalo disease diagnosis system based on BP neural network [5]. However, the above attempts mostly target only one disease and cannot transfer the model from one disease to another.

Previous studies have also developed dialogue systems to interact with patients and give more personalized medical instructions [6]. Xu et al., proposed a method called KR-DQN that embedded entity relations into a DQN agent [7]. To achieve better reuse of the model across different diseases, Qianlong Liu et al., proposed a task-oriented computer-aided diagnosis system based on reinforcement learning. The agent collects information about the symptoms by continuously interacting with the patient to achieve the recognition of several different diseases [8]. Similar to their task, some works developed symptom checkers for online healthcare services by reinforcement learning, using DQN methods to inference diagnosis [9]. However, a single reinforcement learning model is not friendly for the recognition of multiple diseases. As the number of diseases increases, reinforcement learning will suffer from dimensional disasters. With the application of hierarchical reinforcement learning in course recommendation [10], relation extraction [11], and visual dialogue [12], it has been successfully demonstrated that hierarchical reinforcement learning can solve the dimensional catastrophe problem to some extent. Inspired by this, Qianlong Liu et al., proposed a computer-aided diagnosis system based on a hierarchical reinforcement learning model, which solved the problem of plummeting quasi-accuracy rate of disease judgments as the number of diseases and disease symptoms grows [13].

Although the proposed hierarchical reinforcement learning model performs better than the original model, we believe that there are other methods to further alleviate the problem of dimensional disaster. In our opinion, previous attempts all ignore the mining of disease and disease symptom information. After getting the patient’s symptom information, these models simply mark the slot values corresponding to the symptoms and simply select the next action for the agent by BP neural network. However, we believe that mining for disease and disease symptom information is essential, and experiments have shown that the accuracy of the model’s judgment of disease can be greatly improved by modeling symptom information. Adequate consideration of various factors in data mining can ensure mining efficiency and quality [14]. Our goal is to find a better way of processing and mining disease symptom information for the original model.

Filters are an important part of signal processing, which allows specific frequency components of the signal to pass through while greatly attenuating other frequency components. Using the frequency selection function of the filter, a pure signal can be obtained after filtering out noise [15]. The computer-aided diagnosis process also has the problem of “noise”. Considering the real situation, a patient with hyperthyroidism may have symptoms of both hyperthyroidism and fever, but hyperthyroidism is more likely to lead the model to make a correct judgment of hyperthyroidism, while fever is a common symptom of many diseases, which may lead the model to make an “arbitrary” judgment in the diagnosis of the disease. Therefore, fever looks like a “noise” in the information transmission. We believe that instead of giving the same weight to all symptoms, a filter-like approach should be used to leave typical symptoms such as excessive thyroid hormone and filter out the “symptom noise” that occurs in many diseases.

Inspired by these points, this paper proposes two methods for processing disease symptom information. First, similar to the role of filters in signal transmission, we designed filters for disease symptoms, which can effectively filter out the “symptom noise” that reduces the accuracy of model judgment, further remove useless information and reduce the dimension of the agent.

Second, each symptom is represented by a three-dimensional vector ([1,0,0], [0,1,0], [0,0,1] subscales for yes, no, and indeterminate), and each response is considered as a channel. Three channels form the first category of channels; the other considers a disease symptom as a channel, and the set of all symptoms forms the second category of channels. We modeled the two types of channels separately to obtain the importance of different channels in their respective categories and to achieve the goal of distinguishing the importance of symptoms. Experiments demonstrate that both approaches help the model to improve the accuracy of disease judgments.

## 2. Hierarchical Reinforcement Model

### 2.1. Markov Decision Process

Reinforcement learning is an interactive learning technique by interacting with users and obtaining feedback [16]. In recent years, reinforcement learning has made great breakthroughs in many complex decision-making problems [17]. It mainly consists of an intelligent agent and the environment that interact through three signals: state, action, and reward. Assuming that $X\left(t\right)$, *t* is a random process, and the random variable $X\left(t\right)$ has known variables $X\left(t_{1}\right)=x_{1}$, $X\left(t_{2}\right)=x_{2}$, …, the conditional distribution function is only related to $X\left(t_{n}\right)=x_{n}$, but not related to $X\left(t_{1}\right)=x_{1}$, …, $X\left(t_{n-1}\right)=x_{n-1}$. This is the the Markov condition. A reinforcement learning task is called a Markov Decision Process (MDP) if it satisfies the Markov property [18].

A Markov decision process is a sequential decision process that can be represented by a quadruple [19]:(1)$$<S,A,P_{ss^{\prime }}^{a},R_{ss^{\prime }}^{a}>$$
where *S* is a set of states describing the system environment [20]. *A* is the set of all discrete actions an agent can choose from [20]. $P_{ss^{\prime }}^{a}$ represents the probability that a state may be reached $s^{\prime }$ in the next step, after giving it a status *s* and an action *a*. $R_{ss^{\prime }}^{a}$ represents the expected value obtained after reaching the state $s^{\prime }$ given the state $s^{\prime }$ and action *a*.

In any reinforcement learning model, the goal of an intelligent agent is to find a suitable strategy to maximize the rewards from the environment. However, the rewards obtained by the agent in the current state include all possible rewards in the future. To make the intelligence more focused on the present and reduce the influence of the future, a discount factor γ must be introduced into the action-value function to attenuate the influence of future uncertainties.
(2)$$Q^{\pi }\left(s,a\right)=E\left(R_{t}|s_{t}=s,a_{t}=a\right)=E_{\pi }\left[\sum_{k=0}^{\infty }\gamma^{k}r^{t+k+1}|s_{t}=s,a_{t}=a\right]$$

The optimal action-value function follows the Bellman equation:(3)$$Q^{\pi }\left(s,a\right)=E_{s^{\prime }}\left[r+\gamma maxQ^{*}\left(s^{\prime },a^{\prime }|s_{t}=s,a_{t}=a\right)\right]$$

Only if for each state and action follow $Q^{\pi }\left(s,a\right)=Q^{*}\left(s,a\right)$, then this strategy is optimal.

### 2.2. Hierarchical Reinforcement Learning Model

Although reinforcement learning has a strong learning ability, it has its weaknesses and gradually appears in its continuous development. Hierarchical reinforcement learning can solve the problem of dimensional disaster to a certain extent, which makes it show more excellent processing capabilities in environments with more complex environments and larger action spaces [21]. However, we believe that reinforcement learning coupled with feature processing can take the model performance to the next level. Using the idea of breaking up the whole into parts, our experiment consists of a main-decision maker, a disease classifier, and multiple sub-decision makers. All diseases are divided into nine disease groups according to medical knowledge in advance, and one sub-decision maker is responsible for one disease group.

In this experiment, the main-decision maker’s state is $s_{t}=\left[b_{1},b_{2},\ldots ,b_{n}\right]$. Each vector *b* represents a symptom on the data set, which is a three-dimensional vector representing the user’s response to this symptom (yes, no, not sure). For example, if the user confirms the presence of one symptom, then it will be represented as [1,0,0]. Before the sub-decision maker interacts with the environment, we will extract the symptom information from the disease group that the sub-decision maker is responsible for, to form a new state *s*.
(4)$$s=ExtractState[b_{1},b_{2},\ldots ,b_{n}]$$

The main-decision maker is responsible for the activation of sub-decisions makers and disease classifiers. Its action space is $A^{m}=\left\{a^{i}|i=1,2,\ldots ,m\right\}$, *m* is the sum of the sub-decisions and the disease classifier, indicating that at a certain moment *t*, the main-decision maker activates one of the sub-decisions to collect more symptom information or activates the disease classifier to make the final inference.

The action space of the sub-decision maker is $A^{n}=\left\{a^{i}|i=1,2,\ldots ,n\right\}$, and *n* is the sum of all symptoms in their respective groupings. At the current time *t*, the sub-decision generates an action $a_{i}$, asking the patient whether the symptom indexed as *i* is present or not, based on the current state *s*.

The disease classifier is not responsible for interacting with the environment. After being activated by the main-decision maker, the disease classifier makes a final disease identification judgment based on the total disease symptom information provided by the main-decision maker.

Before each diagnosis begins, the model receives a dominant symptom from the outside world to mimic the user’s first elaboration of his or her dominant symptom. The model then continuously interacts with the environment to get the patient’s implicit symptoms until a final identification is made.

## 3. Disease Symptom Information Filter

Based on the hierarchical reinforcement learning model, we design corresponding disease symptom filters for each decision-maker, helping them focus on important disease symptom information. We first analyzed each group and calculated the frequency *f* of each symptom of various diseases in this group.
(5)$$f_{i}=\sum_{i=1}^{n}if\left(a_{i}=true\right)/m$$

Where *m* represents the total number of people suffering from the disease, *n* represents the total number of current symptoms, $a_{i}=true$ means that the current patient suffers from this symptom.

Considering two situations, one is that because the patients are not professional medical workers, they may make wrong judgments about symptoms such as “abdominal pain” and “stomach pain”; another is that there may be only a very small probability of a certain symptom when suffering from a certain disease. Both of them have the potential to cause very infrequent symptoms, bringing the dimensional disaster to this situation. Adding action space for these is not worthwhile. We will first weed out these symptomatic features.

Based on the analysis of the probability of symptoms appearing in different diseases, we found that there may be a high degree of overlap between the symptoms of different diseases in the same disease group. In the Figure 1, we present the symptom overlap for the first group of diseases (where different colors represent different diseases, and the horizontal axis represents the 28 symptoms with the highest probability of occurrence in this group):

### 3.1. Sub-Decisions Maker Symptom Filter

The goal of the sub-decisions maker symptom filter is to filter out the set of symptoms that are representative of the disease. When the model makes inferential predictions, we believe that the model should focus more on symptom information that has a low overlap rate and appropriately ignore overly repetitive symptom information. If a symptom occurs frequently in different groups of patients with different diseases, its information value will be greatly reduced, even can be regarded as a noise that affects judgment. On the contrary, if a symptom is almost appearing exclusively in one disease, its information value will be greatly enhanced.

For this purpose, we established an n-dimensional disease symptom filter. We regard symptoms with a frequency greater than 0.2 as high-frequency symptoms of a disease, and symptoms below 0.2 are low-frequency symptoms. Symptoms that are only high-frequency symptoms of a disease are considered as the most valuable information and are marked as 1. These disease signature filters will be applied in the future.

### 3.2. Main-Decision Maker Symptom Filter

The goal of the main-decision maker symptom filter is to filter out the set of symptoms that are representative of different disease groups. Similar to the previous approach, for each disease group we calculate the probability that each symptom will occur on this disease group.
(6)$$f_{j}=\sum_{i=1}^{n}if\left(a_{i}=true\right)/m_{j}\left(1\leq j\leq 9\right)$$

Where $m_{j}$ denotes the total number of people in disease group *j*, *n* denotes the sum of all symptoms in group *j*, and $a_{i}=true$ means that the current patient does suffer from this symptom.

We regarded symptoms with a frequency greater than 0.1 as high-frequency symptoms in the total disease characteristics, and those with a frequency below 0.1 as low-frequency symptoms. If a high-frequency symptom is not just a high-frequency feature of a group, then it will be removed from the high-frequency symptom group. We believe that the remaining high-frequency symptoms can be representative of their disease group. We set the dimension of the main-decision maker symptom filter as the sum of the total disease symptoms, and high-frequency features will be given higher weight by multiplying their frequency by 10. This symptom filter will also be applied in the follow-up.

The disadvantage of the disease symptom filter is that it causes some rare symptoms not to be mentioned, making it more difficult to determine the disease in a very small number of patients. However, such a sacrifice is worth it in terms of the improvement in model accuracy.

## 4. Weighted Network

In addition to symptom information filtering, we also perform deeper information mining on the state. Many previous works have been done to improve the performance of the network from the spatial dimension. For example, the Inception network structure embeds multi-scale information and aggregates features from different receptive fields to obtain performance gains [22], while the Inside-Outside network considers spatial context information [23]. Unlike the above approaches, SENet targets the connections between feature channels, the goal is to explicitly the interdependence between channels to improve the quality of network representation [24]. Specifically, the weight of each feature channel is automatically obtained through learning, then enhances useful features and suppresses useless features by these weights, to achieve channel adaptive calibration. Kaolin Jiang et al., also started from the channel and improved the detection accuracy of malicious codes using multi-channel image deep learning [25].

### 4.1. Weighted Network Module

We build a weighted network with the goal of modeling the interdependencies between feature channels to mine the depth information of states. We divide the channels into two categories, one is the three channels consisting of “yes”, “no” and “not sure”, which can be considered as a description of a symptom, similar to the RGB color description of a pixel. The second is to combine all symptom sets into another channel. We model the two channels separately to improve the judgment accuracy of the model.

### 4.2. Channel Modeling and Weighted Network

First, convert the state $s_{1}$ into $c_{1}*h_{1}$ and $c_{2}*h_{2}$, where $c_{1}=3$, $c_{2}$ is the sum of the symptoms that the current decision-maker is responsible for. The two states are, respectively, passed into two weighted network modules. These two weighted networks have no essential difference in structure. Each network is divided into two parts: compression and excitation. Convolutional neural networks are good at extracting local features and downsampling [26], however, it has no global receptive field. The compression part uses a one-dimensional adaptive average pooling operation to convert the input state into $c_{1}*1$ and $c_{2}*1$, the compressed state can be regarded as obtaining a global view on a channel, with the receptive field enlarged. The excitation part consists of two one-dimensional convolutions, one is to compress the channel, the other is to restore the channel to the original number, and finally calculate the weight value of each channel.

The weights are obtained after $s_{1}$ and $s_{2}$ pass through the weighted network. Then, two states are each multiplied by their channel weight matrix. After that, the dimensions are uniformly transformed into 3∗h and finally added. Where *h* represents the sum of symptoms for which the current decision-maker is responsible. The calculation formula is expressed as:(7)$$s=\left(s_{1}*w_{1}+s_{1}\right)+transpose\left(s_{2}*w_{2}+s_{2}\right)$$

Its process is shown in the Figure 2:

Before passing the state *s* into the final single-layer neural network, we need to perform a final noise filter on it. Whether the noise filter belongs to the main-decision maker or the sub-decision maker, they are all one-dimensional vectors, and the dimension is equal to the dimension *h* of the current state *s*. The same index position represents the same disease symptoms. We use the dot product method to expand the valuable disease symptoms in the state *s*, and set the disease symptoms with serious overlap to 0. The calculation formula is:(8)$$s_{i,j}^{\prime }=\left\{\begin{array}{c} 0\left(j\neq n,0\leq j\leq h,0\leq n\leq h\right)\\ s_{i,j}*filter_{n}\left(0\leq j=n<h,0\leq i<3\right) \end{array}\right.\;$$

To avoid the filtered state being too sparse, the original state needs to be superimposed. The specific implementation is shown in the Figure 3:

## 5. Experiments and Results

### 5.1. Data Set

The experimental data comes from a synthetic data set provided by Fudan University. This synthetic data set was constructed based on the disease-symptom database SymCat. The database SymCat contains 801 diseases, which the experimenters divided into t21 groups according to medical norms. They select the most representative 9 groups, each of which contains 10 typical diseases.

Because of the confidentiality of patient visit information in the real world, the experimenters constructed a synthetic data set containing 30,000 visits. Experimenters obtained the probabilities of symptoms associated with each disease from Centers for Disease Control and Prevention (CDC) database. Based on the probability distribution, records were generated for each target disease. Given a disease, the labels of symptoms are sampled correctly or incorrectly, and one of the correct symptoms is randomly selected as the dominant trait and the rest as recessive symptoms. Overall, 80% of the data are used for training and 20% are used for testing.

### 5.2. Main Parameters Settings

For the external environment bonus, the sub-decision maker will receive a +1 bonus for each correct symptom, otherwise only a −1 bonus, and an additional +44 bonus if the final identification is correct, and a −44 bonus if the identification fails or if the maximum number of conversation rounds is reached and still no inference is made. The main-decision maker controls the maximum number of dialogue turns only 25 times. Each sub-decision maker, when activated, has a maximum of 5 conversations at a time, otherwise, it will be forcibly stopped. The high-frequency symptom threshold of the sub-decisions maker is 0.2, and the high-frequency threshold of the main decider is 0.1. The discount factor is set to 0.95, the learning rate is set to 0.0005, and the neural network hidden layer is 512.

### 5.3. Experimental Results and Comparison

We conduct experiments on laboratory equipment. Specifically, the python version we use is 3.7.6 and the pyTorch version is 1.7.1. It is noteworthy that the experiments are conducted on lab’s server with 3.50 GHz CPU and 24220MiB GPU. On the dataset of 30,000 examples, it takes about two days to train.

To demonstrate the improvement of both symptom filtering and weighted network for disease diagnosis, we set up a total of five comparison trials. They are hierarchical reinforcement model, hierarchical reinforcement learning model plus symptom filtering, hierarchical reinforcement model plus weighted network, hierarchical reinforcement learning model plus symptom filtering and weighted network, and SVM. We will pass all the correct symptoms to the SVM classification model at one time, and it does not need to obtain any disease symptom information from the patients, so the recognition accuracy of the SVM model is regarded as the upper limit based on the hierarchical reinforcement learning model. The experimental results are in Table 1 and Figure 4:

To prove that our model can effectively alleviate the problem of the curse of dimensionality, we also conduct another comparative experiment. We use three methods on different sizes of the collection, including SVM, hierarchical reinforcement learning (HRL), and hierarchical reinforcement learning model plus symptom filtering and weight calculation. It can be seen from the figure that with the increase of the data set, the accuracy of HRL drops much faster than our model. Additionally, SVM does not need agents, so it is minimally affected. From Figure 5, we can also learn that the most difficult problem in improving the accuracy of the model is how to get more correct symptoms.

In order to determine whether the differences in the results are definitely due to the algorithm’s performance and not due to random algorithmic procedures, we set a *t*-test, which can test whether the difference between two groups is significant. The independent sample *t*-test statistic can be expressed as:(9)$$t=\frac{\bar{X_{1}}+\bar{X_{2}}}{\sqrt{\frac{\left(n_{1}-1\right){S_{1}}^{2}+\left(n_{2}\right){S_{2}}^{2}}{n_{1}+n_{2}-2}\left(\frac{1}{n_{1}}+\frac{1}{n_{2}}\right)}}$$

Where $S_{1}$, $S_{2}$ are the sample variances, and $n_{1}$, $n_{2}$ are the sample sizes.

We conduct five experiments on our model and the original model separately, assuming they are not significantly different, and setting α=0.05. We use the scipy library to do the calculation and get *p* value = 3.91 × 10${}^{-11}$. So, we can conclude that our algorithm can effectively improve the accuracy of the model.

Additionally, to prove that our method is not only effective on the current dataset, we also compare it with more existing methods on two other datasets, the Muzhi dataset, and the Dxy dataset [27].

The Muzhi dataset is collected from the pediatric department on Chinese online healthcare website. It contains 710 user goals and 66 symptoms, with four kinds of labeled diseases, including upper respiratory infection, children’s functional dyspepsia, infantile diarrhea, and children’s bronchitis. The Dxy Dialogue dataset is collected from another Chinese online healthcare community. This dataset contains 527 user goals and 41 symptoms.

On these two datasets, we make a comparison with some existing methods.

Flat-DQN: It has one layer policy and an action space including both symptoms and diseases.

HRL-pretrained: It has two levels, where the low level policy is pre-trained first and then the high level policy is trained.

DQN+relation branch: It has one layer, but added the technology of relation branch.

The results can be seen from Table 2 and Table 3.

Due to the low number of diseases and symptoms in the datasets, the accuracy of all models improved while the number of turns was greatly reduced. Additionally, our model still has a good performance in terms of accuracy. We believe that the advantage of our model would be more pronounced if tested on a large dataset.

Because real-world patient information is confidential, it is difficult to find large symptom-disease datasets. In the future, we may choose another way, just like the dataset we used provided by Fudan University. This synthetic dataset was constructed based on the symptom–disease database called SymCat, which contains 801 diseases. If it is difficult to obtain diseases and symptoms from patients, we can manually label the corresponding symptoms for various diseases according to the disease databases, and establish a new symptom–disease database, which can be used for experiments in medical diagnosis.

## 6. Discussion and Conclusions

In this paper, we mainly further processed the disease symptom information collected by the agent in reinforcement learning. In hierarchical reinforcement learning, every agent has an action space, after receiving an external reward, it will make the next choice. In order to help the agent make a better decision, we did some methodological research. The methods adopted are symptom filtering and weight calculation, one for disease symptom overlap, and another for state channel modeling. The experimental results demonstrate the effectiveness of these two methods in improving the accuracy of the hierarchical reinforcement learning model. In the task-oriented dialogue system, it is the first attempt to design filters to filter the collected information to obtain more critical information, and its combination with weight calculation reflects the effect that one plus one is greater than two.

In the future, we will conduct more in-depth research on two aspects. First, we hope to establish a reasonable disease knowledge base, which is a knowledge base for medical professionals, while taking into account the general public [28], introducing more diseases from the real world, making the knowledge base more complete. In addition to diseases and disease knowledge, symptom filters for disease groups should also be part of the knowledge base, which can be dynamically updated. Second, hierarchical reinforcement learning does not fully solve the problem of dimensional disaster caused by the exponential growth of action space and state space [29], so we will try to make these two methods general, which can help other models based on hierarchical reinforcement learning make some progress on their tasks.

## Figures and Tables

**Figure 1 entropy-24-00931-f001:**
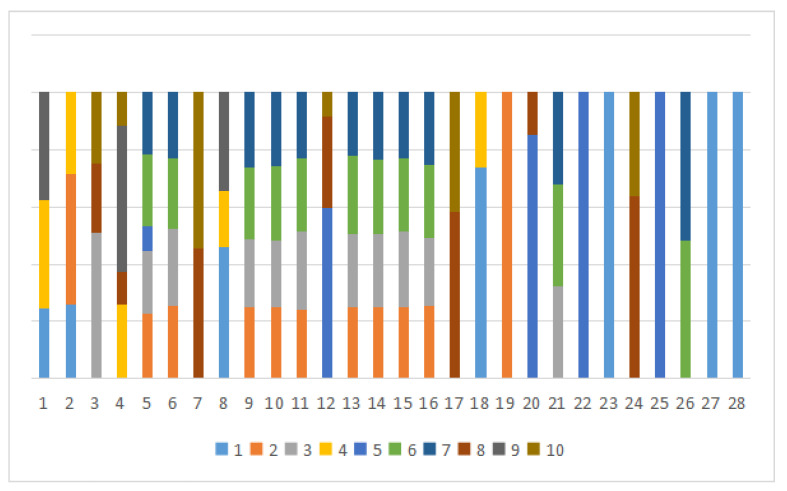
The *x*-axis represents the 28 symptoms with the highest probability, and the *y*-axis represents the proportion. Each term describes the disease distribution for a given symptom.

**Figure 2 entropy-24-00931-f002:**
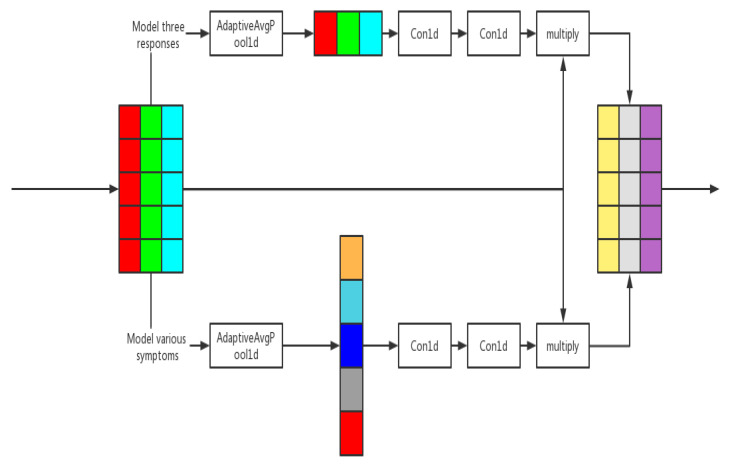
Weighted network module.

**Figure 3 entropy-24-00931-f003:**
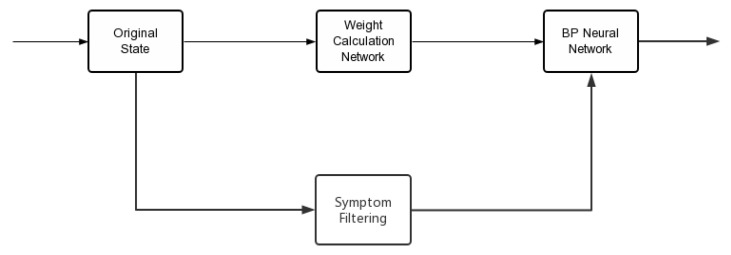
Status processing flow.

**Figure 4 entropy-24-00931-f004:**
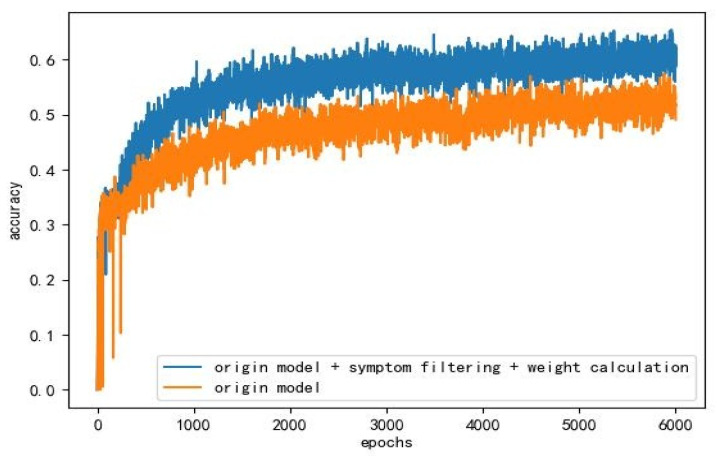
The specificcomparison between the improved hierarchical reinforcement model and the original model in 6000 rounds of training.

**Figure 5 entropy-24-00931-f005:**
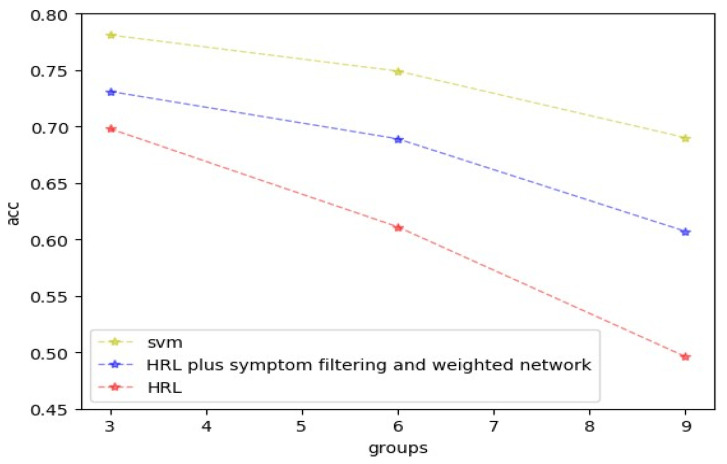
Three methods on different sizes of the collection (SVM, HRL, HRL plus symptom filtering and weight calculation).

**Table 1 entropy-24-00931-t001:** The performance of each model on the data set.

Model	Accuracy	Award	Average Epochs
HRL	49.5%	0.473	16.2
HRL plus Symptom filtering	52.9%	0.61	19.76
HRL plus Weighted network	53.4%	0.76	20.18
Our model	60.5%	2.13	19.23
SVM	69.8%	/	/

**Table 2 entropy-24-00931-t002:** The performance of each model on the Muzhi dataset.

Method	Accuracy	Award	Average Epochs
Flat-DQN	68.2%	0.513	2.712
HRL-pretrained	69.9%	0.563	2.815
DQN+relation branch	70.1%	0.595	2.651
Our model	71.0%	0.613	2.893

**Table 3 entropy-24-00931-t003:** The performance of each model on the Dxy dataset.

Model	Accuracy	Award	Average Epochs
Flat-DQN	68.5%	0.522	2.720
HRL-pretrained	69.7%	0.559	2.800
DQN+relation branch	70.1%	0.578	2.649
Our model	71.9%	0.580	2.805

## Data Availability

All data included in this study are available upon request by contacting the corresponding author.

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
