# Peer review of "Research on Computer-Aided Diagnosis Method Based on Symptom Filtering and Weighted Network"

_entropy, 2022, doi:10.3390/e24070931_

Round 1

Reviewer 1 Report

- Please include the Markov condition in 2.1

  • In page 4, in paragraph 3 , clarify the paragraph starting with " Considering..."
  • In 5.1 what is the meaning of t21?
  • In 6 you mention agents.  Please clarify how do you use them
  • 25% of your references are 5 or more years old.

Reviewer 2 Report

The paper proposes symptom filtering and a weighted network in order to improve the judgment accuracy of the hierarchical reinforcement model in diagnostic tasks. Having gone through the manuscript this reviewer has the following main concerns. 

The research is focused on a single database from which it is impossible to extend the drawn conclusions or suggest that the approach is valid for all diagnostic tasks in general. Possibly, the paper title, solution, and in general the paper may be valid only for the database under test and so the paper should be focused explicitly. 

Another main concern from this reviewer is the fact that evaluation lacks the means to determine the model performance. This is an important drawback that makes the reader question the technical soundness of the proposed approach.

A major concern from this reviewer is the lack of significance tests to determine whether the differences in the results are definitely due to the algorithm's performance (e.g. normality tests, parametric/nonparametric statistical significance tests, etc). 

Finally, a fair comparison of the proposed models with other databases is mandatory to suggest that the approach is a generic computer-aided diagnosis method as the title suggests. 

Reviewer 3 Report

The paper presents a methods of using symptom filtering and a weighted network with the goal of improving the  process of disease identification from a collection set, as the number of diseases increases and consequentely the size of the collection increases. Paper claims that the feature extraction methods proposed in the paper can help existing models improve their accuracy by more than 10%. Here are my main comments:

- On the part of literature review, increase the literature review with more works. Also how you differ from existing works make it more clear

- On the experimental side:

   - you may use the same method on different size of the collection. In this way you can also show the improvements as depending on collection size

 - you may have a discussion on which kind of equipment you did the experiment? How much time did it take to do the analysis? Can your software be used for a medical recommender

  - make a better comparison with existing methods

with other words develop more on the analysis itself

Round 2

Reviewer 1 Report

The issues pointed were properly addresses and the work can continue its editorial process

Author Response

Point 1: The issues pointed were properly addresses and the work can continue its editorial process

Response 1: Thanks for your advice. I added a significance test to determine whether the differences in the results are definitely due to the algorithms performance and not due to random algorithmic procedures. You can see it in page 9. I also introduced the experimental equipment, etc. You can see it in page 8 and page 10.

Reviewer 2 Report

This reviewer acknowledges the manuscript has increased its technical content and some comments made in the original review have been addressed. Nevertheless, the following major technical concern still applies to the resubmitted version:

A major concern from this reviewer is still the lack of significance tests to determine whether the differences in the results are definitely due to the algorithm's performance and not due to random algorithmic procedures and/or data set peculiarities. Statistical significance tests (e.g. normality tests, parametric/nonparametric statistical significance tests, etc) will ensure the approach`s performance is exactly due to the new algorithm's design and whose results and benefits can be reproduced later with other datasets. This is an important issue, and it is even more important now that the authors have included more datasets. From this reviewer's viewpoint, statistical significance tests are mandatory considering the nature of the sets and the datasets evaluated. 

Author Response

Point 1: A major concern from this reviewer is still the lack of significance tests to determine whether the differences in the results are definitely due to the algorithm's performance and not due to random algorithmic procedures and/or data set peculiarities. Statistical significance tests (e.g. normality tests, parametric/nonparametric statistical significance tests, etc) will ensure the approach`s performance is exactly due to the new algorithm's design and whose results and benefits can be reproduced later with other datasets. This is an important issue, and it is even more important now that the authors have included more datasets. From this reviewer's viewpoint, statistical significance tests are mandatory considering the nature of the sets and the datasets evaluated. 

 Response 1: Thanks for your advice. Now I set a significance test to determine whether the differences in the results are definitely due to the algorithms performance and not due to random algorithmic procedures. We use T-Test, which is used to determine whether there is a significant difference between the mean values of two groups of data. We conduct five experiments on our model and the original model separately, assuming they are not significantly different, and setting . We use the scipy library to do the calculation and get . So we can conclude that our algorithm can effectively improve the accuracy of the model. You can see more details in page 9.

Reviewer 3 Report

I think that the current version is an improvement of the previous one. However, not all comments are addressed:

you may have a discussion on which kind of equipment you did the experiment? How much time did it take to do the analysis? Can your software be used for a medical recommender.

Response 3: I think this method has the opportunity to be applied to medical auxiliary diagnosis. The problem is that the patient’s situation is confidential, how can we obtain as much real case data as possible, and professional personnel are required to make reasonable groupings of diseases in advance.

- First which kind of equipment was used to test the AI engine with its performances should be explained in the text. Some discussion on complexity should be also done

- Second, it is said that data size is a problem and how to obtain it is another problem because of ethics, confidentiality, etc. This is a general problem in medical IT. There should be some discussion in the article regarding how this effort should be used in real life practice because this is an applied problem.

Author Response

Point 1:  First which kind of equipment was used to test the AI engine with its performances should be explained in the text. Some discussion on complexity should be also done

Response 1: I have machines on the lab’s server for testing and experimentation.  Specifically, the python version I am using is 3.7.6 and the pyTorch version is 1.7.1. It is noteworthy that the numerical simulations are conducted on lab’s server with 3.50 GHz CPU and 24220MiB GPU.

About how much time it takes to analyze,  it is mainly determined by the dialogue rounds and the size of the dataset. We set the maximum number of  rounds to be 25. On a dataset of 30,000 cases, it took more than two days to complete the data analysis, while on the dataset of about 500examples, it only takes about 10 hours.  

Point 2: Second, it is said that data size is a problem and how to obtain it is another problem because of ethics, confidentiality, etc. This is a general problem in medical IT. There should be some discussion in the article regarding how this effort should be used in real life practice because this is an applied problem.

Response 2: Because real-world patient information is confidential, it’s difficult to find large symptom-disease datasets. In the future, we may choose another way, just like the dataset we used provided by Fudan University. This synthetic dataset was constructed based on the symptom-disease database called SymCat, which contains 801 diseases. If its difficult to obtain diseases and symptoms from patient, we can manually label the corresponding symptoms for various diseases according to the disease databases, and establish a new  symptom-disease database, which can be used for experiments in medical diagnosis.

You can see more details in page 8 and page 10.

Round 3

Reviewer 2 Report

The authors have addressed this reviewer's main concerns. The paper is now acceptable. 

Reviewer 3 Report

Authors answered the comments of the reviewer. I will give an accept